# The probability of reducing hospitalization rates for bronchiolitis with epinephrine and dexamethasone: A Bayesian analysis

Larry Dong[1,2*], Terry P. Klassen[3], David W. Johnson[4,5], Rhonda Correll[6], Serge Gouin[7,8], Maala Bhatt[9,10], Hema Patel[11,12], Gary Joubert[13], Karen J. L. Black[14], Troy W. S. Turner[15,16], Sandra R. Whitehouse[17], Amy C. Plint[10,18☙], Anna Heath[1,2,19☙]

**1** Division of Biostatistics, Dalla Lana School of Public Health, University of Toronto, Toronto, Canada, **2** Child Health Evaluative Sciences, Hospital for Sick Children, Toronto, Canada, **3** Children's Hospital Research Institute of Manitoba, Department of Pediatrics and Child Health, Rady Faculty of Health Sciences, University of Manitoba, Winnipeg, Manitoba, Canada, **4** Alberta Children's Hospital Research Institute, Calgary, Canada, **5** Departments of Pediatrics, Emergency Medicine and Physiology & Pharmacology, Cumming School of Medicine, University of Calgary, Calgary, Canada, **6** Children's Hospital of Eastern Ontario Research Institute, Ottawa, Canada, **7** Pediatric Emergency Department, Centre Hospitalier Universitaire Sainte-Justine, Montréal, Canada, **8** Department of Pediatrics, Université de Montréal, Montréal, Canada, **9** Department of Pediatrics, University of Ottawa, Ottawa, Canada, **10** Children's Hospital of Eastern Ontario, Ottawa, Canada, **11** McGill University Health Centre Research Institute, Montréal, Canada, **12** Department of Pediatrics, McGill University, Montreal, Canada, **13** Department of Pediatrics, Children's Hospital, Western University, London, Canada, **14** Division of Pediatric Emergency Medicine, British Columbia Children's Hospital, University of British Columbia, Vancouver, British Columbia, Canada, **15** Pediatric Emergency Department, Stollery Children's Hospital; Alberta, Canada, **16** Department of Pediatrics, University of Alberta. Edmonton, Alberta, Canada, **17** Department of Pediatrics, University of British Columbia, Vancouver, Canada, **18** Department of Pediatrics and Emergency Medicine, University of Ottawa, Ottawa, Canada, **19** Department of Statistical Science, University College London, London, United Kingdom

☙ Contributed equally as co-senior authors.
* larry.dong@mail.utoronto.ca

## Abstract

### Background

Bronchiolitis exerts a high burden on children, their families and the healthcare system. The Canadian Bronchiolitis Epinephrine Steroid Trial (CanBEST) assessed whether administering epinephrine alone, dexamethasone alone, or in combination (EpiDex) could reduce bronchiolitis-related hospitalizations among children less than 12 months of age compared to placebo. CanBEST demonstrated a statistically significant reduction in 7-day hospitalization risk with EpiDex in an unadjusted analysis but not after adjustment.

### Objective

To explore the probability that EpiDex results in a reduction in hospitalizations using Bayesian methods.

**Data availability statement:** Data cannot be shared publicly because participants of this study did not agree for their data to be shared publicly. Data are available from Amy C. Plint

(plint@cheo.on.ca) or the CHEO research institute (researchdatamanagement@cheo.on.ca) for researchers who meet the criteria for access to confidential data.

**Funding:** This work was supported by Canada Research Chairs in Statistical Trial Design [AH] and Clinical Trials [TPK] (https://www.chairs-chaires.gc.ca/home-accueil-eng.aspx), Tier 1 University of Ottawa Research Chair [ACP] (https://www.uottawa.ca/facultymedicine/research-and-innovation/research-chairs) and an operating grant from Canadian Institutes of Health Research (CIHR; https://cihr-irsc.gc.ca/e/193.html) for the original CanBEST study. The funders plated no role in the study design, data collection and analysis, decision to publish, or preparation of the manuscript.

**Competing interests:** Amy C. Plint receives in-kind support from Amphastar for a clinical trial in bronchiolitis. The other authors have no relevant conflicts to disclose.

## Study design

Using prior distributions that represent varying levels of preexisting enthusiasm or skepticism, i.e., how confident or doubtful one is that EpiDex may reduce hospitalizations, and information about the treatment effect *before* data were collected, the posterior distribution of the relative risk of hospitalization compared to placebo was determined. The probability that the treatment effect is less than 1, 0.9, 0.8 and 0.6, indicating increasing reductions in hospitalization risk, are computed alongside 95% credible intervals.

## Results

Combining a minimally informative prior distribution with the data from CanBEST provides comparable results to the original analysis. Unless strongly skeptical views about the effectiveness of EpiDex were considered, the 95% credible interval for the treatment effect lies below 1, indicating a reduction in hospitalizations. There is a 90% probability that EpiDex results in a clinically meaningful reduction in hospitalization of 10% even when incorporating skeptical views, with a 67% probability when considering strongly skeptical views.

## Conclusion

A Bayesian analysis demonstrates a high chance that EpiDex reduces hospitalization rates for bronchiolitis, although strongly skeptical individuals may require additional evidence to change practice.

## Trial registration

**Clinical Trial registry name, registration number**: Current Controlled Trials number, ISRCTN56745572

---

## Introduction

Bronchiolitis is a respiratory disease that exerts significant burden on the healthcare system [1]. It is the leading cause of infant hospitalization in North America and is associated with substantial healthcare spending during the winter months [2–5]. Few treatments that have conclusively demonstrated a reduction in hospitalization rates for infants with bronchiolitis [6,7]. The Canadian Bronchiolitis Epinephrine Steroid Trial (CanBEST) is one of the largest trials in bronchiolitis and examined the effectiveness of epinephrine, dexamethasone, and their combination in reducing the risk of hospitalization by day 7 in children aged 6 weeks to 12 months of age [8]. CanBEST used a factorial design [9] to randomize participants to one of four treatment categories: a combination of nebulized epinephrine and oral dexamethasone (EpiDex), nebulized epinephrine with oral placebo (Epi), oral dexamethasone with nebulized placebo (Dex) and oral and nebulized placebo (placebo). This design

allowed CanBEST to evaluate Epi, Dex and EpiDex to determine whether any of these three intervention arms resulted in a reduction in hospitalization compared to placebo.

CanBEST demonstrated a clinically meaningful 35% reduction in the relative risk (RR) of hospitalization (a 9.3% absolute risk reduction) for EpiDex compared to placebo [8] and used the standard statistical *frequentist approach* to draw conclusions from the study. The frequentist approach for statistical analysis indirectly evaluates study hypotheses by calculating the chance of observing the available data under the assumption that a null hypothesis is true [10], typically that there is no effect of treatment on the outcome of interest. The frequentist approach assumes that if the chance of observing the data when the null hypothesis is true is small, then the null hypothesis can be rejected in favor of an alternative hypothesis, usually that there is a beneficial treatment effect. Within this framework, answering multiple research questions within the same study, as for the CanBEST study, usually requires an adjustment to maintain appropriate error rates [11]. However, there is a controversy around whether this is required when testing for interactions in a factorial design, such as CanBEST [9,12]. As a result, the CanBEST study presented both an unadjusted and adjusted analysis. The unadjusted analysis resulted in a statistically significant reduction in hospitalization with EpiDex at the 5% level with a p-value of 0.02, while the analysis adjusted for multiple comparisons was not statistically significant with a p-value of 0.07 [8]. The discrepancy has led to challenges in interpreting and translating the results of the CanBEST study to the bedside [13] and currently, national guidelines for bronchiolitis recommend only supportive care for patients with bronchiolitis [6]. However, extensive basic science literature demonstrates that co-administration of beta$_2$-adrenoceptor agonists and corticosteroids mutually enhance each other's effectiveness [14–18] and their synergy is also well documented in clinical trials of asthma management [19,20].

An alternative approach to frequentist statistical analysis [21] is also available and gaining popularity: the Bayesian approach [22,23]. This framework allows you to calculate the probability that an intervention is effective, given the observed data [24,25]. The Bayesian approach also incorporates pre-existing evidence or clinical expertise into the statistical analysis [26] and can thus examine how differences in clinical judgment and experience of an intervention affect the interpretation of results [27]. Finally, as Bayesian analyses are only dependent on the data collected, the proposed model and the prior distributions [28], we circumvent multiple testing issues [9]. Given the extensive health system, patient and family burden of bronchiolitis and the lack of recommended interventions to reduce this burden [1], we undertake an unplanned Bayesian analysis of the data from the CanBEST study. This analysis will calculate the probability that EpiDex reduces hospitalizations for bronchiolitis [27].

## Methods

### Canadian Bronchiolitis Epinephrine Steroid Trial

CanBEST was a multicenter, double-blinded placebo-controlled clinical trial that assessed the efficacy of epinephrine and dexamethasone, alone and in combination, each compared to placebo, as a treatment for children aged 6 weeks to 12 months who presented at the emergency department with bronchiolitis [8]. All hospitals who participated in CanBEST are members of the national research network, Pediatric Emergency Research Canada (PERC). Children's Hospital of Eastern Ontario Research Ethics Board gave approval in May 2004 for the CanBEST study with REB Number 02/59E. CanBEST recruited between 1st December 2004 and 31st March 2008. Written informed consent was obtained from the parents or guardians of all infants included in the study. The primary outcome was admission to hospital for bronchiolitis within seven days of study enrolment. The complete inclusion and exclusion criteria, outcome definitions and study procedures are provided in the primary publication [8]. Trial participants were randomized equally into one of the four treatment groups: EpiDex, Epi, Dex or placebo, with dosing details provided in the primary publication [8]. The target enrolment was 800 patients. Three participants were lost to follow-up meaning data for 797 participants were available for this post-hoc analysis.

## An introduction to Bayesian inference

Bayesian and frequentist methods for statistical analysis differ in their philosophy, leading to differences in their conduct and interpretation [26]. Frequentist analyses reach statistical conclusions by controlling error rates over many analyses conducted in the same manner [29]. When multiple research questions are evaluated within the same study, frequentist reasoning clarifies that the chance of at least one incorrect conclusion is increased and necessitates adjustments to control the error rate of the overall study [11]. In contrast, Bayesian methods aim to make the *best* conclusions using the data from the specific study [25] meaning that study conclusions depend on the *data* and assumed model, rather than the analysis method [25].

To perform a Bayesian analysis, a *prior distribution* is required that represents the available evidence, usually assumed to derive from the literature or relevant clinical experience, about the plausible range of the treatment effect *before* analyzing the data [30]. This prior distribution is combined with the study data to determine an updated probability distribution for the treatment effect, which then represents our knowledge about the plausible values of the treatment effect. This is known as *posterior* distribution. From this, we can determine the probability that the intervention is beneficial. This probability is not available from frequentist p-values [24] and allows us to trade-off the chance of experiencing benefit or harm from an intervention [31].

## Design of prior distributions: reference priors and data-driven priors

Designing prior distributions is a crucial element of undertaking a Bayesian analysis. Furthermore, it is an inherently subjective process, which has led to criticisms of Bayesian statistics as prior distributions influence the trial analysis [32]. However, we exploit this by selecting a range of prior distributions that explicitly represent different archetypes of beliefs about the efficacy of the interventions and results from previously conducted studies [27]. This allows us to explore how variations in the views on the effectiveness of EpiDex influence the interpretation of the CanBEST study. This is advantageous for two reasons; firstly, given the pre-existing controversy on the efficacy of EpiDex for bronchiolitis, we can gain further insight into the debate around its effectiveness. Secondly, readers can determine which prior best represents their own background assessment of the efficacy of EpiDex, based on their experience and expertise, and interpret the results of the CanBEST study accordingly [27].

We consider two classes of prior distributions in our analysis: reference priors and data-driven priors [27]. Reference priors represent clinical archetypes of beliefs about the treatment effectiveness: strongly enthusiastic, moderately enthusiastic, moderately skeptical, strongly skeptical and no opinion. The "no opinion" option uses a "minimally informative" prior to exert the smallest possible influence on the results and provides a similar numerical result to a frequentist analysis but allowing for a Bayesian interpretation. The other four reference priors are defined to reflect the level of enthusiasm or skepticism about the effect of EpiDex using a normal distribution for the log-RR (Fig 1) [27]. Generally, we assume that skeptics believe there is no treatment effect (corresponding to a treatment effect of 1) while enthusiasts believe the treatment is effective at reducing hospitalization (a treatment effect less than 1). Table 1 presents the five reference priors for treatment effect for EpiDex. As there is no credible evidence that these therapies would increase hospitalization, we did not consider this in our reference priors.

In addition to the reference priors, we used *data-driven* priors, which were derived by fitting a mixed effects hierarchical model using uninformative priors and data from previously published randomized trials in bronchiolitis [27,33]. Broadly, the results using the data-driven priors can be interpreted as combining data from the CanBEST study with previous studies, like a meta-analysis. Studies deemed sufficiently close to CanBEST were chosen according to the four following criteria: (a) study participants were randomized to either a glucocorticoid steroid, a $\beta_2$-agonist or their combination; (b) the outcome of interest was related to hospital admission – ideally 7-day cumulative hospitalization, (c) participants were infants less than two years of age and (d) the study was conducted prior to the publication of the initial CanBEST analysis. These inclusion criteria assume that drugs in the same class as dexamethasone and epinephrine will have similar effectiveness.

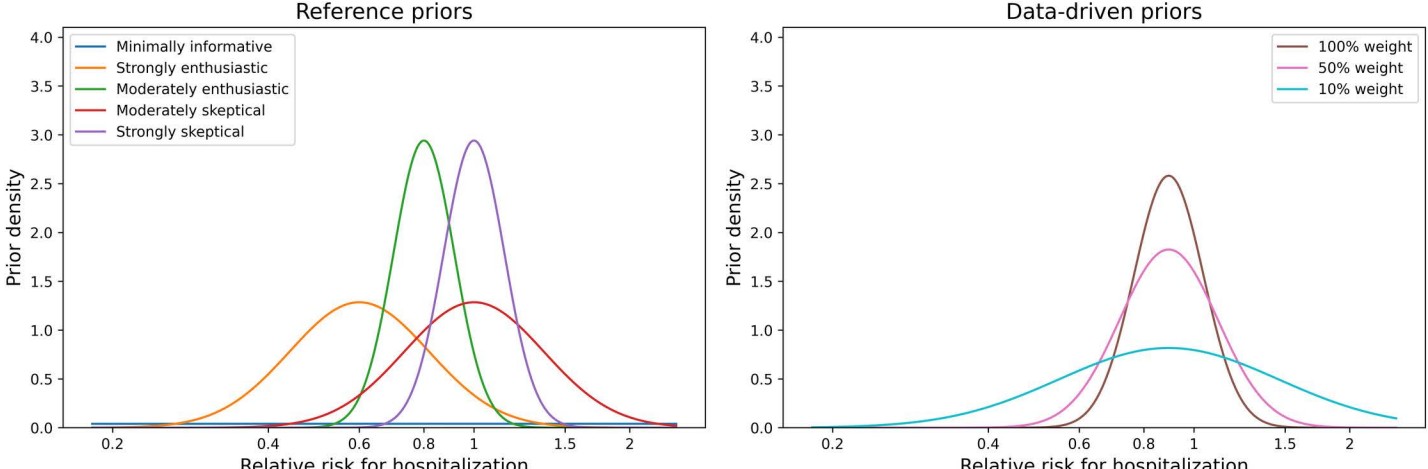

**Fig 1.** *Left: Reference priors for the relative risk of hospitalization within 7 days after treatment administration. Right: Data-driven priors using different weights to control the influence from previous studies.*

**Table 1.** *The prior distributions used for the CanBEST reanalysis.*

| Prior Belief | Median relative risk of hospitalization with EpiDex | SD of log relative risk | Equivalent Prior Sample Size | Prior probability that the relative risk is below threshold (%) | | | |
|---|---|---|---|---|---|---|---|
| | | | | RR<1 | RR<0.9 | RR<0.8 | RR<0.6 |
| **Reference Priors** | | | | | | | |
| Minimally Informative | 1 | 10 | ≈ 0 | 50 | 50 | 49 | 48 |
| Strongly Enthusiastic | 0.6 | 0.31 | 134 | 95 | 90 | 82 | 50 |
| Moderately Enthusiastic | 0.8 | 0.14 | 668 | 94 | 80 | 50 | 2 |
| Moderately Skeptical | 1 | 0.31 | 125 | 50 | 37 | 24 | 5 |
| Strongly Skeptical | 1 | 0.14 | 594 | 50 | 23 | 6 | ≈ 0 |
| **Data-Driven Priors** | | | | | | | |
| 100% Weighting | 0.89 | 0.15 | 478 | 77 | 52 | 23 | ≈ 0 |
| 50% Weighting | 0.89 | 0.21 | 272 | 70 | 51 | 30 | 3 |
| 10% Weighting | 0.89 | 0.47 | 61 | 59 | 51 | 41 | 20 |

We chose an age range of participants up to 24 months to reflect the variation among clinicians and guidelines in defining the age range of children that may be deemed to have bronchiolitis [7]. Table A.1 in Appendix A in S1 File summarizes the intervention, comparator, population, outcome of interest, relative risk, and sample size for studies used for the data-driven priors. There were differences in the chosen studies, including variation in the choice of drug, its dosing, patient inclusion/exclusion criteria and outcomes. Prior sample sizes were calculated by equating the product of treatment effect variances, prior and posterior, with their corresponding sample sizes [34].

Once the data-driven priors have been specified (Fig 1), we consider scenarios that dilute their impact on the final analysis. These scenarios are based on providing a prior "weight" of 100%, 50% and 10%, which represents the relative contribution of a participant in a previous study compared to the contribution of a participant in the CanBEST study and is controlled by the standard deviation of the prior [34]. This weighting procedure – applied on the variance of treatment effect estimates in the hierarchical mixed effects model – accounts for fundamental differences between the data in CanBEST and the data in the previous studies, such as differences in patient population, interventions, and outcomes

   

of interest. Thus, data-driven priors are rarely developed using a full systematic review and meta-analysis as the down-weighting allows us to "discount" the contribution of studies that do not entirely match the CanBEST study.

Table 1 provides a descriptive summary of all the considered reference and data-driven priors for EpiDex. We report the median RR of hospitalization, the standard deviation (SD) of the log-RR distribution – where smaller SDs imply more certainty about the treatment effect before seeing the data – and the probability of RR being below various thresholds, e.g., P(RR < 0.9) is the probability that the treatment reduces the probability of hospitalization probability by at least 10%. For similar tables pertaining to Epi alone and Dex alone priors, see Tables B.1 and B.2 in the Appendix B in S1 File.

## Analysis

For each of the prior distributions defined in Table 1, we used a Bayesian model to determine the posterior distributions of the treatment effect for Epi, Dex and EpiDex compared to placebo. We used a binomial generalized linear model with a log link to calculate the relative risk of hospitalization for the three interventions, compared to placebo, adjusted for site. The adjustment for participating sites was achieved using a hierarchical model. The Bayesian model was fitted using PyMC version 4.4.0 [35] in Python version 3.9.15 with 20,000 simulations and a burn-in of 10,000 to ensure convergence [36]. Traceplots were examined to check for convergence and autocorrelation [36].

The posterior distributions were summarized using the median relative risk and equi-tailed 95% credible intervals; these quantities are analogues to a frequentist point estimate of effect and confidence interval. Finally, we estimated the probability that the relative risk was below the thresholds 1, 0.9, 0.8 and 0.6 by the proportion of the simulations that were below each of those thresholds. These thresholds represent a reduction in the risk of hospitalization for bronchiolitis of 0%, 10%, 20% and 40%, respectively.

## Results

The primary outcome was available for 797 infants, of these 34 who received the EpiDex combination, 47 who received Epi alone, 51 who received Dex alone and 53 who received placebo were admitted to hospital for bronchiolitis within 7 days of study enrolment.

Overall, our Bayesian analysis (Table 2) demonstrated a posterior probability that the use of EpiDex results in a reduction in hospitalization greater than 98%, unless the clinician was strongly skeptical about the effectiveness of EpiDex. This means that there is over a 98% probability that using EpiDex to treat bronchiolitis in infants in the ED makes them less

Table 2. *Summary table of group 1 (EpiDex) posterior characteristics: median RR, 95% credible interval and probability of RR smaller than various thresholds.*

| | Posterior median for the RR of hospitalization (95% Credible Interval) | Posterior probability that the RR of hospitalization is below threshold (%) | | | |
|---|---|---|---|---|---|
| | | RR < 1 | RR < 0.9 | RR < 0.8 | RR < 0.6 |
| **Reference Priors** | | | | | |
| Minimally informative | 0.66 (0.45, 0.96) | 99 | 95 | 84 | 30 |
| Strongly enthusiastic | 0.63 (0.45, 0.85) | 100 | 99 | 94 | 39 |
| Moderately enthusiastic | 0.74 (0.60, 0.91) | 100 | 97 | 77 | 3 |
| Moderately skeptical | 0.75 (0.55, 1.00) | 98 | 89 | 67 | 8 |
| Strongly skeptical | 0.86 (0.70, 1.06) | 92 | 65 | 23 | ≈ 0 |
| **Data-Driven Priors** | | | | | |
| 100% weighting | 0.77 (0.62, 0.96) | 99 | 92 | 63 | 1 |
| 50% weighting | 0.74 (0.56, 0.96) | 99 | 93 | 73 | 7 |
| 10% weighting | 0.69 (0.48, 0.96) | 99 | 94 | 82 | 23 |

likely to be hospitalized. The complete results from our Bayesian analysis are displayed in Table 2, while the equivalent analyses for the Epi and Dex treatment groups are available in Tables B.1 and B.2 in the Appendix B in S1 File. Posterior distributions for the RR of EpiDex compared to placebo are displayed in Fig 2, with reference priors on the left and data-driven priors on the right.

### Reference priors

All equi-tailed 95% credible intervals exclude a null relative risk value of 1 except when using a strongly skeptical prior. Using minimally informative priors for all treatment effects leads to an estimated posterior median RR of 0.66 for EpiDex and a corresponding 95% equi-tailed credible interval of (0.45, 0.96); this result is similar to the initial CanBEST analysis where the estimated RR and 95% confidence interval were 0.65 and (0.45, 0.96), respectively, in the unadjusted analysis. Comparing estimates for the posterior median, we can see that the RR increases as the prior skepticism increases. Finally, the probability of a reduction in hospitalization rates with EpiDex, compared to placebo (RR < 1) is greater than 98%, unless a strongly skeptical prior is used. Similarly, the probability of a greater than 10% reduction in hospitalization rates is greater than 90%, unless individuals are strongly skeptical.

### Data-driven priors

For the data-driven priors, all equi-tailed 95% credible intervals exclude a null relative risk value of 1, indicating that the combined current evidence suggests a reduction in hospitalization rates with EpiDex. Increasing the weighting of the previous studies increases the posterior median RR, from 0.69 to 0.77, indicating that the previous studies demonstrated a smaller treatment effect than the effect observed in CanBEST. We confirm this trend by computing the posterior distributions using data-driven priors with increasing weights between 0% and 100%; this analysis is available in Appendix D in S1 File.

## Discussion

Bronchiolitis exerts a huge burden on the healthcare system, patients, and families [1]. Our Bayesian analysis of the results from the pivotal CanBEST trial has demonstrated that there is a greater than 98% probability that EpiDex reduces hospitalizations for bronchiolitis compared to placebo unless clinicians are highly skeptical. Even highly skeptical

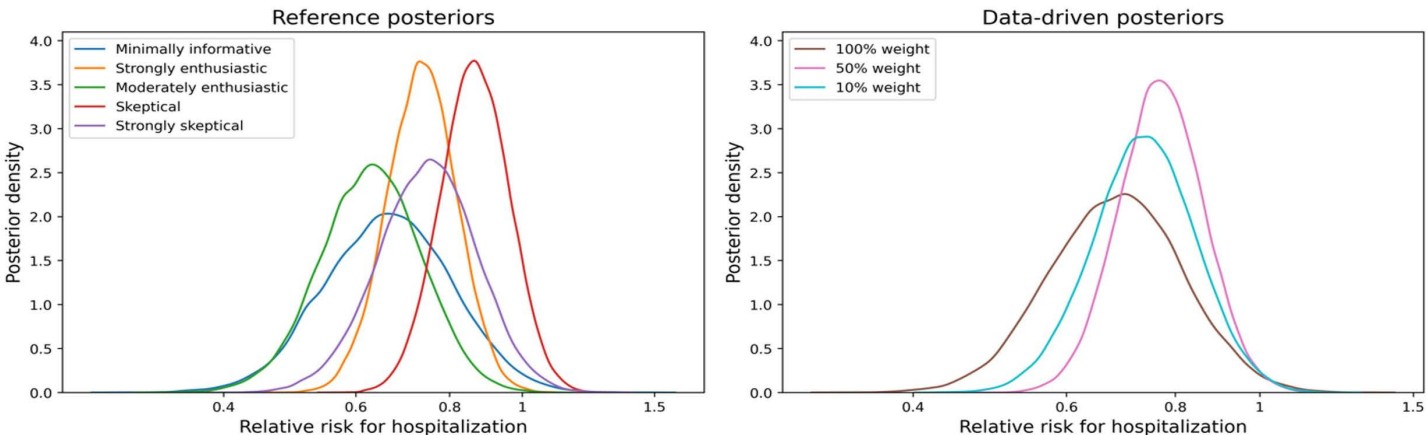

**Fig 2.** *Posterior distributions for the relative risk of hospitalization within 7 days after treatment under different reference priors (left) and data-driven priors (right).*

individuals could be swayed by the data in the CanBEST study as our analysis demonstrates that there is a 90% chance that EpiDex reduces hospitalizations. This finding was also supported when we combined CanBEST with data from previous studies.

Furthermore, our data-driven Bayesian analysis confirmed that CanBEST resulted in a larger reduction in hospitalization rates compared to previous studies as the estimated posterior median RR for EpiDex increases as the weight for the prior studies increases. Overall, we conclude that EpiDex has the potential to reduce admissions to hospital for bronchiolitis and, as a result, the burden of bronchiolitis for infants, their families, and the healthcare system.

The estimated posterior probability of a 10% reduction in the relative risk of hospitalization varies between 65%, for the strongly skeptical prior, to 99%, for the strongly enthusiastic prior and is always greater than 90% for the data-driven priors. This demonstrates not only does EpiDex potentially reduce hospitalizations, but also there is a relatively high chance of a *clinically meaningful* reduction in hospitalization rates with EpiDex. However, if there is high skepticism about the efficacy of EpiDex, additional evidence may be required before being convinced by the outcome of CanBEST. In contrast, even when moderate skepticism is considered, the CanBEST results demonstrate a probability of clinically meaningful reduction in hospitalization rates of 89%.

Our Bayesian reanalysis has added important nuance to the interpretation of CanBEST. Firstly, we calculated the probability that the interventions are effective at reducing the risk of hospitalization, which is not possible in standard analyses. This can facilitate conversation and aligns with how clinicians make decisions when deciding on patient care [24]. Secondly, by representing a wide spectrum of prior beliefs, we have provided a flexible framework for interpreting the CanBEST results, facilitating discussion between clinical decision makers who may have differing experience and expertise. That being said, it can be argued that, within the range of priors that we have considered, minimally informative and data-driven priors represent impartiality and can be considered most appropriate. Finally, the design of priors and the lack of a strict definition for statistical significance in the Bayesian paradigm encourages an in-depth discussion of the implications of the results from CanBEST and whether EpiDex can be used to alleviate the overwhelming health system impact of bronchiolitis, particularly in face of the recent bronchiolitis surges [37].

There are some limitations to this reanalysis. Firstly, any inherent limitations in CanBEST are not circumvented by this analysis [27]. Bayesian methods provide a different framework for the interpretation and dissemination of results but are unable to overcome challenges in the design of the initial trial. For example, the definition of bronchiolitis varies globally and CanBEST restricted participants to infants less than one year of age who were experiencing wheezing for the first time in the typical "season" for respiratory syncytial virus infection. Other jurisdictions include children up to 24 months of age and do not always restrict the diagnosis to those with a first episode of wheezing [38]. Clinicians may also be concerned about the use of corticosteroids in young children although a recent comprehensive systematic review found no increased risk of short–term adverse effects among children with acute respiratory illnesses treated with corticosteroids compared to placebo [39]. Similarly, no trial of nebulized epinephrine in bronchiolitis has demonstrated serious side effects or clinically important increases in heart rate or blood pressure. A theoretical risk is that children treated with epinephrine and discharged home might clinically worsen as the effect of epinephrine wears off. However, a systematic review of bronchiolitis studies found similar return-to-care rates in children treated with epinephrine as compared with placebo and salbutamol [40]. Strengths of the original trial are also inherent to this analysis, e.g., the CanBEST trial has very limited loss to follow up. A limitation of this analysis is that alternative prior distributions could have been considered and would have changed the results [26]. However, by being explicit about our prior definitions and assumptions and considering a range of previous studies for the data-driven priors, we allow the reader to determine which view and analysis aligns most closely with their beliefs. Lastly, this re-analysis uses data from CanBEST, wherein data was collected between 2004 and 2008. Post-COVID bronchiolitis and its seasonality may have changed, with recent studies highlighting an increased severity of the disease, risk of hospitalization and potential benefit of RSV vaccination as a preventive measure [41,42]. Furthermore, with the advent of RSV monoclonal antibody use among a wider population of infants (e.g., term, healthy

infants), the risk of hospitalization from bronchiolitis should be reduced and possibly reduce the need for a change in management [43].

## Conclusion

Bayesian analysis provides an alternative to the commonly used frequentist interpretation of clinical trials. It allows individuals with different prior experience and expertise to contextualize their interpretation of the trial results. For CanBEST, our Bayesian analysis demonstrated a very high probability that the combination of nebulized epinephrine and oral dexamethasone reduces bronchiolitis-related hospital admissions. Thus, use of this combination treatment is likely to reduce the substantial burden of bronchiolitis to both infants and their families and the healthcare system. The use of Bayesian methods circumvents a discussion on whether the analysis should be adjusted for multiple comparisons, which previously complicated the interpretation of CanBEST. Note that due to the uncertain interpretation of CanBEST in the frequentist paradigm, an international randomized trial is currently underway to answer the calls for further evidence on the effectiveness of the combined therapy [44]. Based on our analysis, the results from this international trial will be particularly relevant to clinicians that are highly skeptical about the effectiveness of the combined therapy.

## Supporting information

**S1 File. Supplementary results from the analysis and description of the data-driven priors.**
(DOCX)

## Acknowledgments

This results from this analysis were presented at the Statistical Society of Canada Conference 2023.

## Author contributions

**Conceptualization:** Amy C. Plint, Anna Heath.

**Data curation:** Terry P. Klassen, David W. Johnson, Rhonda Correll, Serge Gouin, Maala Bhatt, Hema Patel, Gary Joubert, Karen J. L. Black, Troy W. S. Turner, Sandra R. Whitehouse, Amy C. Plint.

**Formal analysis:** Larry Dong.

**Funding acquisition:** Amy C. Plint.

**Methodology:** Larry Dong, Anna Heath.

**Supervision:** Amy C. Plint, Anna Heath.

**Visualization:** Larry Dong.

**Writing – original draft:** Larry Dong, Amy C. Plint, Anna Heath.

**Writing – review & editing:** Terry P. Klassen, David W. Johnson, Rhonda Correll, Serge Gouin, Maala Bhatt, Hema Patel, Gary Joubert, Karen J. L. Black, Troy W. S. Turner, Sandra R. Whitehouse.

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
