## [Decision Letter · Decision Letter 0]

10 Feb 2025

PONE-D-24-16700The Probability of Reducing Hospitalization Rates for Bronchiolitis: A Bayesian AnalysisPLOS ONE

Dear Dr. Heath,

Thank you for submitting your manuscript to PLOS ONE. After careful consideration, we feel that it has merit but does not fully meet PLOS ONE’s publication criteria as it currently stands. Therefore, we invite you to submit a revised version of the manuscript that addresses the points raised during the review process.

The manuscript has been evaluated by two reviewers, and their comments are available below.

The reviewers have raised a number of concerns that need attention. They request improvements to the reporting of methodological and statistical aspects of the study, and improvements to the writing and discussion.

Could you please revise the manuscript to carefully address the concerns raised?

We look forward to receiving your revised manuscript.

Kind regards,

Helen Howard

Staff Editor

PLOS ONE

Reviewers' comments:

Reviewer's Responses to Questions

**Comments to the Author**

1. Is the manuscript technically sound, and do the data support the conclusions?

Reviewer #1: Yes

Reviewer #2: Partly

2. Has the statistical analysis been performed appropriately and rigorously? 

Reviewer #1: I Don't Know

Reviewer #2: No

3. Have the authors made all data underlying the findings in their manuscript fully available?

Reviewer #1: Yes

Reviewer #2: No

4. Is the manuscript presented in an intelligible fashion and written in standard English?

Reviewer #1: Yes

Reviewer #2: Yes

5. Review Comments to the Author

Reviewer #1: Thank you for asking me to review your paper on bronchiolitis. I agree that it will take a further study with strong evidence to convince clinicians to start using epidex for treatment of patients with bronchiolitis. All the evidence to date and guidelines recommends minimal handling and not to use any medications. I feel the paper explains clearly about the Can BEST study and explains the results of the original paper and the reasoning for re-analysis using the Bayesian method. I have to say that I don’t fully understand the statistics and have recommended the paper be reviewed by a statistician. The paper is heavy in statistical methods and discussion so I agree that it will not convince many frontline clinicians to adjust their practise and start using epidex in the management of bronchiolitis. I think it is a valuable debate to have about the methods used for analysis of RCTs but would generally err on supporting simple methods that reflect a pragmatic approach.

It will be interesting to see the results of the BIPED study and then see if the combined data shows that epi and dex truly reduces the hospitalisation of children with bronchiolitis.

Major changes:

I think there needs to be an acknowledgement that the data for this re-analysis is from 2004-2008 when bronchiolitis was different to what we now see. Post covid the bronchiolitis season has dramatically changed and we are only just starting to see a return to normality. It maybe that emerging bronchiolitis studies demonstrate changes in the natural course of the disease.

I think the limitations of this re-analysis needs to further develop the potential risks of using dex and epi in children with bronchiolitis. The importance of safe prescribing can not be ignored and we know that the current management of infants with bronchiolitis with minimal intervention is safe and causing little harm.

I would be keen to see if there is any difference when you assess the different risk groups such as those with underlying conditions or born extreme preterm.

It is also important to note that with the introduction of RSV vaccinations the rates of bronchiolitis admissions to hospital should be significantly reduced. This may therefore negate the need for changing in our clinical management of bronchiolitis as it is hoped the burden from bronchiolitis will be significantly reduced.

Minor changes:

I would suggest changing the title to include epinephrine and steroids. Maybe ‘The probability of reducing hospitalisation rates for bronchiolitis with epidex using a Bayesian analysis method.’

In the design of prior distributions section the third criteria for suitable studies close to CanBEST is participants were infants less than two years of age but Can BEST was based on less than 12 months of age. I feel this would lead to significant differences in the cohorts of children being included as the many of the children 1-2 years of age will have a different aetiology to their respiratory illness other than bronchiolitis. This will therefore impact on the results of those studies and so I would assume impact the data driven priors. As stated I am not a statistician and so unsure how you would take into account the difference in ages in the different studies and into the Bayesian analysis model.

In the discussion section it is stated ‘Our Bayesian analysis of the results from the pivotal CanBEST trial has demonstrated that there is a greater than 98% probability that EpiDex reduces hospitalizations for bronchiolitis compared to placebo unless clinicians are highly skeptical.’ which is a repeat of what has been described in the results. I would like to see more detail about what does the greater than 98% probability mean?

In the limitations section the references 38 and 39 are based on evidence from young children or children under 2 years of age. Again I do not feel this is representative of the bronchiolitis group of children who should be defined as less than 12 months of age to avoid confusion with different respiratory illness aetiologies such as episodic wheeze which may be more likely to benefit from steroids or epi. I would also say another limitation of the re-analysis is the unfamiliarity of clinicians with Bayesian analysis and therefore it is unlikely to lead to changes in clinical practise.

Reviewer #2: The aim of this study is to reanalyze data from a previous study that applied frequentist statistics using Bayesian analysis. The study requires a more detailed description of its methodology and results. Another problem is that the present findings lead to uncertain conclusions, depending on the strength of the prior belief. For certain readers, such as policymakers, this uncertainty may not be reassuring. I think there is value in the fact that the focus of the study is on the probability of the treatment effect, however, this should be communicated in an effective manner.

First, there is a mismatch between the presented results in Table 2 and Figure 2. The color of strongly skeptical shows a posterior distribution with highest density for a value around 0,75. The color of skeptical distribution shows the highest density for a value around 0,86. The moderately enthusiastic distribution peaks at something around 0,6. The strongly enthusiastic distribution peaks at around 0,75. Finally, the minimally informative peaks at around 0,66. This is inconsistent with Table 2.

Furthermore, inconsistencies also happen for the results using data-driven priors in Figure 2. The 100% weight appears to have the smallest median value. The 50% weight has the largest median value. The 10% weight has an intermediate median value.

Second, the use of data-driven prior distributions is an interesting approach for the reason pointed out in the manuscript that the analysis considers the data of the present study with prior knowledge of the previous studies. There are two problems. One is the methodology is still not clear on how the weights are applied. The other problem is that the upper bound of the posterior interval remains unchanged regardless of how informative the priors are. For instance, it is difficult to accept no change in the upper bound for the 10% weight distribution.

Third, for a reader that is not used with Bayesian analysis, results that vary depending on the views might be hard to accept. In the absence of previous data, I would suggest the minimally informative prior. IN this case, according to Table 2, the probability of RR below threshold is high. Since there is previous knowledge, the data-driven priors are an interesting approach. However, it is not clear how this was applied to be able to fully judge the results.

Although I see some value in using different degrees of views (skeptical, enthusiastic etc.), I would say that a manuscript with minimally informative, and data-driven only priors would better communicate results. This means all other results not incorporated in the manuscript.

Additional comments below.

Abstract

The term Bayesian distribution should be replaced with posterior distribution for accuracy.

Correction: The probability that the treatment effect is less than 1, 0.9, 0.8 and 0.6

For a reader of the abstract, the meaning of skeptical views is not clear. It may be helpful to introduce this concept in the background section. The same applies to strongly skeptical individuals. After re-reading the manuscript, I understand what is meant, but the abstract should be accessible to a general audience.

Introduction

Statement about 9.3% absolute risk reduction (line 16): is this from ref 8?

Methods

I do not think the description of frequentist methods in line 64 is fair and should be revised.

"Frequentist analyses reach statistical conclusions by controlling error rates over many analyses conducted in the same manner (29). When multiple research questions are evaluated within the same study, the chance of making at least one incorrect conclusion is increased and necessitates adjustments to control the error rate of the overall study."

The phrase "treatment effect after seeing the data" (line 76) seems very conversational and should be revised.

About the statement:"Secondly, readers can determine which prior best represents their own background assessment of the efficacy of EpiDex, based on their experience and expertise, and interpret the CanBEST study results accordingly." This is a strong claim, implying that the results are subjective, which I do not agree with. It should be reworded to avoid this implication.

Regarding Table 1: Inform which distributions are being used.

It is not well understood how the weighting is applied to data-driven priors. Is it applied to the standard deviation to make the prior less informative? Further clarification on this point would be helpful. Formulas can be useful as well.

Results

For some unknown reason, when I download Figure 1 (JPEG), only Figure 2 is downloaded. I wanted to view Figure 1 in higher resolution.

Figures 1 and 2: The tables use the term moderately skeptical. I recommend maintaining this term in the figures for consistency.

When a probability is found to be 0, it is best to report it as less than a negligible number, such as 0.001.

The previous comments about Figure 2 on the mismatch of the distributions.

How is the equivalent prior sample size calculated? This was not explained in the methodology.

Figure D1: The distribution of RR seems bounded on top by 1. What explains this behavior?

Discussion

Last sentence: should be revised.

6. PLOS authors have the option to publish the peer review history of their article (what does this mean?). If published, this will include your full peer review and any attached files.

Reviewer #1: **Yes: **Martin Edwards

Reviewer #2: No

---

## [Author Response · Author response to Decision Letter 0]

2 Apr 2025

Reviewer #1: Thank you for asking me to review your paper on bronchiolitis. I agree that it will take a further study with strong evidence to convince clinicians to start using epidex for treatment of patients with bronchiolitis. All the evidence to date and guidelines recommends minimal handling and not to use any medications. I feel the paper explains clearly about the Can BEST study and explains the results of the original paper and the reasoning for re-analysis using the Bayesian method. I have to say that I don’t fully understand the statistics and have recommended the paper be reviewed by a statistician. The paper is heavy in statistical methods and discussion so I agree that it will not convince many frontline clinicians to adjust their practise and start using epidex in the management of bronchiolitis. I think it is a valuable debate to have about the methods used for analysis of RCTs but would generally err on supporting simple methods that reflect a pragmatic approach.

It will be interesting to see the results of the BIPED study and then see if the combined data shows that epi and dex truly reduces the hospitalisation of children with bronchiolitis.

Dear Reviewer #1,

Firstly, thank you for taking the time to review our paper and propose changes with the goal to improve it.

Major changes:

I think there needs to be an acknowledgement that the data for this re-analysis is from 2004-2008 when bronchiolitis was different to what we now see. Post covid the bronchiolitis season has dramatically changed and we are only just starting to see a return to normality. It maybe that emerging bronchiolitis studies demonstrate changes in the natural course of the disease.

RESPONSE: While we agree that the seasonality of bronchiolitis changed during the pandemic and is now returning to a more typical seasonality, we are not aware of evidence suggesting changes in the natural history of the disease per se related to COVID-19. However, the use of RSV monoclonal antibody among a wider population of children may influence severity and risk of hospitalization and we have included a statement to that effect in our limitations. As such, we have edited the last sentence of the limitations to read: “Lastly, this re-analysis uses data from CanBEST, wherein data was collected between 2004 and 2008. Post-COVID bronchiolitis and its seasonality may have changed, with recent studies highlighting increased severity and risk of hospitalization, potentially decreasing the relevance of this analysis. Furthermore, with the advent of RSV monoclonal antibody use among a wider population of infants (e.g. term, healthy infants), the risk of hospitalization from bronchiolitis should be reduced and possibly reduce the need for a change in management.”

I think the limitations of this re-analysis needs to further develop the potential risks of using dex and epi in children with bronchiolitis. The importance of safe prescribing can not be ignored and we know that the current management of infants with bronchiolitis with minimal intervention is safe and causing little harm. I would be keen to see if there is any difference when you assess the different risk groups such as those with underlying conditions or born extreme preterm.

RESPONSE: Unfortunately, we do have the data available to investigate these subgroup analyses. Overall only five percent of children in the original CanBEST study had an underlying condition, and children at high risk of severe bronchiolitis (those with underlying significant heart disease, chronic lung disease, and immunodeficiency) were not enrolled in the trial. Only 10% of children in CanBEST were born preterm (defined as less than 37 weeks in the trial). However, we agree that this point would be worth exploring in future research.

It is also important to note that with the introduction of RSV vaccinations the rates of bronchiolitis admissions to hospital should be significantly reduced. This may therefore negate the need for changing in our clinical management of bronchiolitis as it is hoped the burden from bronchiolitis will be significantly reduced.

RESPONSE: We agree that the introduction of RSV monoclonal antibody use for otherwise healthy infants will/has reduced the burden of bronchiolitis and has lessened the risk of hospitalization. We have added this to our limitations as outlined above

Minor changes:

I would suggest changing the title to include epinephrine and steroids. Maybe ‘The probability of reducing hospitalisation rates for bronchiolitis with epidex using a Bayesian analysis method.’

RESPONSE: We have made these changes.

In the design of prior distributions section the third criteria for suitable studies close to CanBEST is participants were infants less than two years of age but Can BEST was based on less than 12 months of age. I feel this would lead to significant differences in the cohorts of children being included as the many of the children 1-2 years of age will have a different aetiology to their respiratory illness other than bronchiolitis. This will therefore impact on the results of those studies and so I would assume impact the data driven priors. As stated I am not a statistician and so unsure how you would take into account the difference in ages in the different studies and into the Bayesian analysis model.

RESPONSE: We chose to include studies with children up to two years of age to reflect the variation in national guidelines and clinician practice in defining the age range of children that may be deemed to have bronchiolitis. However, we acknowledge that the selection of studies for inclusion in the prior is not easy, which is why we decided to use the weighing approach to reduce the impact of the prior on the results. This aims to account for differences between the study populations. We have also added the following sentence when discussing the design of the prior distributions: “These inclusion criteria assume [...] that outcomes in participants between ages one and two are comparable to those in CanBEST.”

In the discussion section it is stated ‘Our Bayesian analysis of the results from the pivotal CanBEST trial has demonstrated that there is a greater than 98% probability that EpiDex reduces hospitalizations for bronchiolitis compared to placebo unless clinicians are highly skeptical.’ which is a repeat of what has been described in the results. I would like to see more detail about what does the greater than 98% probability mean?

RESPONSE: We clarify using the following passage: “this means that there is over a 98% probability that using EpiDex to treat bronchiolitis in infants makes them less likely to be hospitalized”.

In the limitations section the references 38 and 39 are based on evidence from young children or children under 2 years of age. Again I do not feel this is representative of the bronchiolitis group of children who should be defined as less than 12 months of age to avoid confusion with different respiratory illness aetiologies such as episodic wheeze which may be more likely to benefit from steroids or epi.

RESPONSE: As highlighted above, we have included these children to reflect differences in guidelines.

I would also say another limitation of the re-analysis is the unfamiliarity of clinicians with Bayesian analysis and therefore it is unlikely to lead to changes in clinical practise.

RESPONSE: While this may be true, we do not believe that it is an inherent limitation of the study we have performed. In this study, we have aimed to introduce the concept of Bayesian analysis and use a range of priors to explore how the CanBEST analysis is influenced by the choice of prior. We have clarified that the goal of this analysis is to support discussion: “by representing a wide spectrum of prior beliefs, we have provided a flexible framework for interpreting the CanBEST results, facilitating discussion between clinical decision makers who may have differing experience and expertise.”

Reviewer #2:

Dear Reviewer #2, we thank you for your detailed comments on our manuscript. We address each of them below.

The aim of this study is to reanalyze data from a previous study that applied frequentist statistics using Bayesian analysis. The study requires a more detailed description of its methodology and results. Another problem is that the present findings lead to uncertain conclusions, depending on the strength of the prior belief. For certain readers, such as policymakers, this uncertainty may not be reassuring. I think there is value in the fact that the focus of the study is on the probability of the treatment effect, however, this should be communicated in an effective manner.

RESPONSE: Thank you for the comments on our manuscript, we’ve made changes throughout based on these reviewer comments to improve clarity.

First, there is a mismatch between the presented results in Table 2 and Figure 2. The color of strongly skeptical shows a posterior distribution with highest density for a value around 0,75. The color of skeptical distribution shows the highest density for a value around 0,86. The moderately enthusiastic distribution peaks at something around 0,6. The strongly enthusiastic distribution peaks at around 0,75. Finally, the minimally informative peaks at around 0,66. This is inconsistent with Table 2.

RESPONSE: In response to this comment, we discovered an error in labelling our plots; these have been updated. We highlight the changes in color labelling below as “prior strength: color of posterior in previous manuscript version -> color in the revised manuscript”:

Minimally informative: blue -> blue

Strongly enthusiastic: green -> orange

Moderately enthusiastic: orange -> green

Moderately skeptical: purple -> red

Strongly skeptical: red -> purple

In short, we kept the legend in the plots the same and ensured that the labelling is accurate.

Furthermore, inconsistencies also happen for the results using data-driven priors in Figure 2. The 100% weight appears to have the smallest median value. The 50% weight has the largest median value. The 10% weight has an intermediate median value.

RESPONSE: This has also been fixed:

100% weight: pink -> brown

50% weight: cyan -> pink

10% weight: brown -> cyan

We apologize for the confusion.

Second, the use of data-driven prior distributions is an interesting approach for the reason pointed out in the manuscript that the analysis considers the data of the present study with prior knowledge of the previous studies. There are two problems. One is the methodology is still not clear on how the weights are applied. The other problem is that the upper bound of the posterior interval remains unchanged regardless of how informative the priors are. For instance, it is difficult to accept no change in the upper bound for the 10% weight distribution.

RESPONSE: We have added a description and citation to highlight how the weights are applied: “These scenarios are based on providing a prior “weight” of 100%, 50% and 10%, which represents the relative contribution of a participant in a previous study compared to the contribution of a participant in the CanBEST study and is controlled by the standard deviation of the prior (A). This weighting procedure – applied on , which increases the variance of treatment effect estimates in the hierarchical mixed effects model – accounts for fundamental differences between the data in CanBEST and the data in the previous studies, such as differences in patient population, interventions, and outcomes of interest.”

We have also verified our results and confirmed that the upper bound of the CI does not change as we are rounding to 2 decimal places.

Third, for a reader that is not used with Bayesian analysis, results that vary depending on the views might be hard to accept. In the absence of previous data, I would suggest the minimally informative prior. IN this case, according to Table 2, the probability of RR below threshold is high. Since there is previous knowledge, the data-driven priors are an interesting approach. However, it is not clear how this was applied to be able to fully judge the results.

RESPONSE: We have clarified that it is the interpretation of the results that change depending on prior beliefs: “Secondly, readers can determine which prior best represents their own background assessment of the efficacy of EpiDex, based on their experience and expertise, and interpret the results of the CanBEST study results accordingly” We have also clarified how the data-driven priors are incorporated into the analysis, particularly regarding the down-weighting procedure which is “applied on the variance of treatment effect estimates in the hierarchical mixed effects model”.

Although I see some value in using different degrees of views (skeptical, enthusiastic etc.), I would say that a manuscript with minimally informative, and data-driven only priors would better communicate results. This means all other results not incorporated in the manuscript.

RESPONSE: The results of the CANBEST analysis have already been published and, as such, this analysis is not aiming to provide a neutral analysis of the CANBEST results. Clinicians often have views on the efficacy of an intervention based on their occasional use in practice or use in other disease areas. As such, we believe that the priors representing different “degrees of belief” can help support debate around the interpretation of the CANBEST analysis and have decided to keep all analyses. This way the reader can reflect on their prior beliefs and find their own unique interpretation. However, we have added a sentence to clarify that the minimally informative and data driven priors demonstrate impartiality: That being said, it can be argued that, within the range of priors that we have considered, minimally informative and data-driven priors represent impartiality and can be considered most appropriate.

Additional comments below.

Abstract

The term Bayesian distribution should be replaced with posterior distribution for accuracy.

RESPONSE: We made this change.

Correction: The probability that the treatment effect is less than 1, 0.9, 0.8 and 0.6

RESPONSE: We made this change.

For a reader of the abstract, the meaning of skeptical views is not clear. It may be helpful to introduce this concept in the background section. The same applies to strongly skeptical individuals. After re-reading the manuscript, I understand what is meant, but the abstract should be accessible to a general audience.

RESPONSE: We have added the following explanation into the abstract: “Using prior distributions that represent varying levels of preexisting enthusiasm or skepticism, i.e. how confident or doubtful one is that EpiDex may reduce hospitalizations,”

Introduction

Statement about 9.3% absolute risk reduction (line 16): is this from ref 8?

RESPONSE: Yes.

Methods

I do not think the description of frequentist methods in line 64 is fair and should be revised.

RESPONSE: We have revised this to read: "Frequentist analyses reach statistical conclusions by controlling error rates over many analyses conducted in the same manner (29). When multiple research questions are evaluated within the same study, frequentist reasoning clarifies that the chance of at least one incorrect conclusion is increased and necessitates adjustments to control the error rate of the overall study."

The phrase "treatment effect after seeing the data" (line 76) seems very conversational and should be revised.

RESPONSE: We have removed this phrasing and the sentence finishes at “treatment effect.”

About the statement: "Secondly, readers can determine which prior best represents their own background assessment of the efficacy of EpiDex, based on their experience and expertise, and interpret the CanBEST study results accordingly." This is a strong claim, implying that the results are subjective, which I do not agree with. It should be reworded to avoid this implication.

RESPONSE: The choice of a prior distribution is a personal one as we have clarified in the abstract, it represents how confident the individual is about the efficacy of EpiDex before seeing the data. As the amount of data increases, the influence of

---

## [Decision Letter · Decision Letter 1]

16 Apr 2025

The Probability of Reducing Hospitalization Rates for Bronchiolitis with Epinephrine and Dexamethasone: A Bayesian Analysis

PONE-D-24-16700R1

Dear Dr. Heath,

We’re pleased to inform you that your manuscript has been judged scientifically suitable for publication and will be formally accepted for publication once it meets all outstanding technical requirements.

Kind regards,

Dhammika Leshan Wannigama, MD PhD

Academic Editor

PLOS ONE

Additional Editor Comments (optional):

All the questions raised have been adequately addressed, and the study provides critical new insights that could shift how we treat bronchiolitis with steroids.

Reviewers' comments:

Reviewer's Responses to Questions

**Comments to the Author**

1. If the authors have adequately addressed your comments raised in a previous round of review and you feel that this manuscript is now acceptable for publication, you may indicate that here to bypass the “Comments to the Author” section, enter your conflict of interest statement in the “Confidential to Editor” section, and submit your "Accept" recommendation.

Reviewer #2: All comments have been addressed

2. Is the manuscript technically sound, and do the data support the conclusions?

Reviewer #2: Yes

3. Has the statistical analysis been performed appropriately and rigorously? 

Reviewer #2: Yes

4. Have the authors made all data underlying the findings in their manuscript fully available?

Reviewer #2: Yes

5. Is the manuscript presented in an intelligible fashion and written in standard English?

Reviewer #2: Yes

6. Review Comments to the Author

Reviewer #2: All comments have been addressed by the authors.

7. PLOS authors have the option to publish the peer review history of their article (what does this mean?). If published, this will include your full peer review and any attached files.

Reviewer #2: No

---

## [Editor Report · Acceptance letter]

PONE-D-24-16700R1

PLOS ONE

Dear Dr. Heath,

I'm pleased to inform you that your manuscript has been deemed suitable for publication in PLOS ONE. Congratulations! Your manuscript is now being handed over to our production team.

Kind regards,

on behalf of

Dr. Dhammika Leshan Wannigama

Academic Editor

PLOS ONE